## Pregnancy physiology pattern prediction study (4P study): protocol of an observational cohort study collecting vital sign information to inform the development of an accurate centile-based obstetric early warning score

Fiona Kumar,[1] Jude Kemp,[1] Clare Edwards,[1] Rebecca M Pullon,[2] Lise Loerup,[2] Andreas Triantafyllidis,[2] Dario Salvi,[2] Oliver Gibson,[2] Stephen Gerry,[3,4] Lucy H MacKillop,[1] Lionel Tarassenko,[2] Peter J Watkinson[5]

For numbered affiliations see end of article.

**Correspondence to**
Dr Peter J Watkinson;
peter.watkinson@ndcn.ox.ac.uk

## ABSTRACT

**Introduction** Successive confidential enquiries into maternal deaths in the UK have identified an urgent need to develop a national early warning score (EWS) specifically for pregnant or recently pregnant women to aid more timely recognition, referral and treatment of women who are developing life-threatening complications in pregnancy or the puerperium. Although many local EWS are in use in obstetrics, most have been developed heuristically. No current obstetric EWS has defined the thresholds at which an alert should be triggered using evidence-based normal ranges, nor do they reflect the changing physiology that occurs with gestation during pregnancy.

**Methods and analysis** An observational cohort study involving 1000 participants across three UK sites in Oxford, London and Newcastle. Pregnant women will be recruited at approximately 14 weeks' gestation and have their vital signs (heart rate, blood pressure, respiratory rate, oxygen saturation and temperature) measured at 4 to 6-week intervals during pregnancy. Vital signs recorded during labour and delivery will be extracted from hospital records. After delivery, participants will measure and record their own vital signs daily for 2 weeks. During the antenatal and postnatal periods, vital signs will be recorded on an Android tablet computer through a custom software application and transferred via mobile internet connection to a secure database. The data collected will be used to define reference ranges of vital signs across normal pregnancy, labour and the immediate postnatal period. This will inform the design of an evidence-based obstetric EWS.

**Ethics and dissemination** The study has been approved by the NRES committee South East Coast–Brighton and Sussex (14/LO/1312) and is registered with the ISRCTN (10838017). All participants will provide written informed consent and can withdraw from the study at any point. All data collected will be managed anonymously. The findings will be disseminated in international peer-reviewed journals and through research conferences.

### Strengths and limitations of this study

► This will be the first study to describe normal ranges of all five vital signs (blood pressure, heart rate, temperature, respiratory rate and oxygen saturation) gathered contemporaneously across gestation in the antepartum, intrapartum and in the postpartum periods.
► Participants will all be within UK healthcare systems. This may limit translation to other healthcare settings.

## BACKGROUND

Early warning score (EWS) systems for obstetric patients assign increasing scores (or colours) to each individual physiological measurement (blood pressure, heart rate, temperature, respiratory rate and oxygen saturation) as they become more abnormal. If either the sum of scores for several physiological measurements, or a score for a single physiological measurement exceeds set thresholds, an alert is triggered. Successive confidential enquiries into maternal deaths in the UK have highlighted an urgent need to develop a national obstetric EWS to aid the more timely recognition, treatment and referral of women who are developing life-threatening complications of pregnancy.[1–3] This recommendation was strongly supported by a national survey of obstetric anaesthetists.[4] It is acknowledged that recognising physiological deterioration is complicated by the normal changes in maternal physiology that occur both during pregnancy and immediately after delivery.[2]

An evidence-based approach is required for widespread acceptance of a national obstetric EWS system.[5] Recognising normality is the first step to understanding abnormality.[6] To understand normal ranges at different gestational ages of 'normal' pregnancy, the intrapartum and the postpartum periods, large numbers of measurements at each point are required. These will allow normal ranges and their associated centiles throughout pregnancy to be defined in the modern era. It is important that these observations are measured at the same time to allow their interactions to be understood. Longitudinal data recorded from the same patients, where possible with the same equipment, across the entire pregnancy and the first 2 weeks after delivery are particularly important to ensure that step changes between the stages of pregnancy are not missed.

Large amounts of longitudinal data for blood pressure during pregnancy already exist (eg, from the longitudinal AVON[7] and BOSHI[8] studies). However, much of this is now more than two decades old. We found less data for heart rate in pregnancy (eg, refs [9–11]). Relatively few data have been collected during the intrapartum period or for the first 2 weeks postpartum (eg, if postnatal data are collected, this only occurs at one or two occasions several weeks after delivery, such as at 6 weeks [12 13] and 6 months [14]). Furthermore, there are very few data available for temperature [15 16], respiratory rate [17 18] and oxygen saturation [17 19] in pregnancy, intrapartum or postpartum, from which to define modern normal-range centiles. The data that are available suggest clear trends occur during pregnancy which may need to be taken into account to ensure appropriate identification of abnormality. For example, systolic blood pressure appears to change by around 6 mm Hg over the course of pregnancy.[7 8] However, while clear vital sign trends are apparent, the range of expected vital sign values during pregnancy is not known, and has not been able to be determined by synthesis of available data sets.

A centile-based approach to identifying abnormality will allow experts to choose which women should be reviewed, based on strong evidence of how far one or more of their vital sign observations lie from normality. The approach is particularly suited to the analysis of data recorded during pregnancy, as well as intrapartum and postpartum, during which the event rate is relatively low.

We have previously used such an approach for both the recognition of physiological deterioration in hospitalised adults[6] and the maintenance of safety in the aerospace industry.[20]

The primary objective of this study is to develop a database of vital sign measurements from pregnancy, labour and the postpartum period (table 1). Estimates of population distributions and associated centiles can be derived from this database. The secondary objective of this study is to use this information to develop a centile-based EWS system for pregnancy, labour and the postpartum period, which will guide clinical experts to provide the most appropriate care.

## METHODS AND ANALYSIS
We prepared this protocol following the STROBE[21] and SPIRIT[22] guidelines, where appropriate.

### Study design
A UK three-centre longitudinal observational study. The study commenced August 2012. We will continue the study until 1000 women contribute vital sign data and the last participant is 14 days postpartum (estimated completion September 2017). The study will take place in two stages. Stage 1, in a single centre (Oxford) as a substudy of the Interbio-21st study (REC:08/H0606/139).[23] Stage 2 will allow two additional centres and the continuation of the study in the first centre after the completion of the Interbio-21st study. Study procedures will be unchanged between phases.

### Study setting
Women will be approached to take part when they attend the ultrasound department or antenatal clinics at three university hospitals (online supplementary appendix). Data will be collected from antenatal visits, hospital admissions and the community.

### Study population
We aim to recruit 1000 women aged 16 or above, with a singleton pregnancy of less than 20 weeks' gestation, who fall within category 1 of the American Society of Anaesthesiologists' classification of physical status at enrolment ('A normal healthy patient without any clinically

| Table 1 | A summary of the primary and secondary objectives and endpoints | |
|---|---|---|
| | **Objectives** | **Endpoints** |
| Primary | ► To develop a database of vital sign measurements during pregnancy, labour and the postpartum period | ► Estimates of distributions and associated centiles for five vital signs throughout pregnancy, labour and the postpartum period |
| Secondary | ► To develop a centile-based early warning scoring system for pregnancy, labour and the postpartum period | ► Centile-based thresholds for alerting for five vital signs throughout pregnancy, labour and the postpartum period |
| | ► To investigate new patterns within vital sign data in pregnancy | ► Trend analyses of five vital signs and inter-relations between these trends |

important co-morbidity and without a clinically significant past/present medical history'). Gestational age will be determined using a measurement of crown-rump length obtained prior to 14 weeks of gestation.[24] We will exclude women with a known condition expected by the recruiting clinician to alter maternal vital signs. The full inclusion and exclusion criteria are detailed in online supplementary appendix 2.

### Vital sign data collected

Five vital signs will be collected: blood pressure, heart rate, oxygen saturation, temperature and respiratory rate. Blood pressure will be measured with a blood pressure monitor approved for use in pregnancy (Microlife 3BT0-A(2)/WatchBP Home, Microlife, Taipei, Taiwan). Heart rate and oxygen saturation will be measured with a Bluetooth-enabled pulse oximeter (WristOx2 3150, Nonin Medical, Minnesota, USA). Temperature will be measured with a tympanic thermometer (Genius 2, Covidien (Medtronic), Dublin, Ireland). In a subgroup, temperature will be measured with an additional Bluetooth thermometer (Fora IRb 20b, ForaCare, Taipei, Taiwan). The device allows for automatic transfer of readings (thus eliminating the risk of user input errors) but is not commonly used in UK clinical practice. Trained midwives will estimate respiratory rate by observing chest wall movement following a standard operating procedure (SOP). A software application on an Android smartphone will also be used to estimate respiratory rate through the detection of chest wall movements with the inbuilt accelerometer and gyroscope. The equipment used for data collection is shown in figure 1.

Vital sign data will be entered (or in the case of Bluetooth-enabled devices, automatically transmitted) onto a tablet computer (Samsung Galaxy Tab 4.0) and sent to a secure database on the National Health Service (NHS) network using 3G/4G. The tablet computer will run a custom Android application that will guide the user through the measurement sequence. Once sent, the data will be automatically deleted from the tablet. All data will be entered, transferred and stored using a unique study number.

Vital sign data recorded during delivery will be extracted retrospectively from the participants' hospital notes once they have been discharged from hospital. All vital signs recorded on existing obstetric EWS charts, anaesthetic charts and labour charts (partogram) will be entered onto the secure 4P study database for analysis. Examples of the intrapartum case report form are included in online supplementary appendix 3.

### Other data collected

In addition, relevant medical and obstetric history will be extracted from the participants' notes to support analysis of the data. We will collect demographic information (age, height, weight, ethnicity, number of previous pregnancies, smoking status), medical and obstetric history, current health status, pregnancy-related health

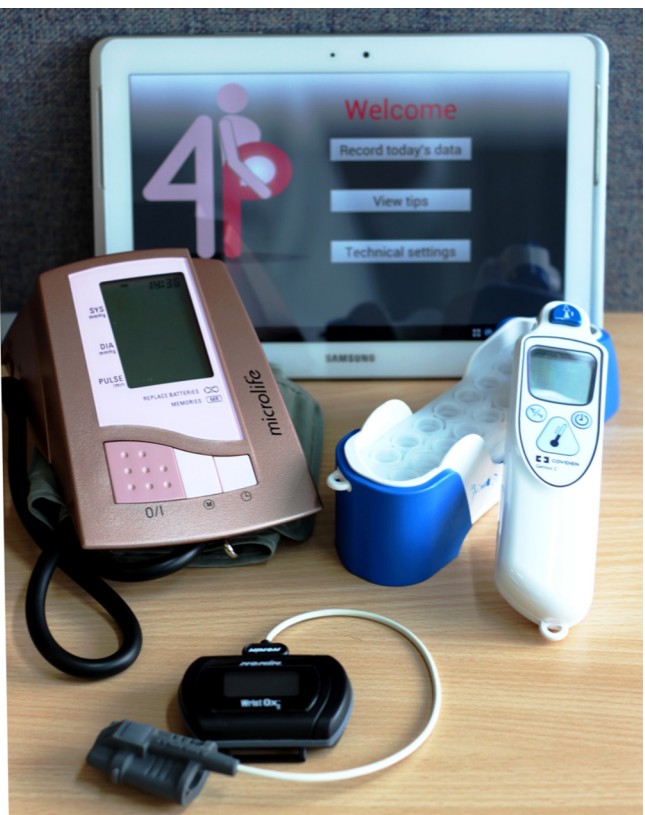

**Figure 1** Equipment used for vital sign data collection.

and current medications at the initial assessment. At each follow-up appointment we will collect assessments of smoking status, current health status, pregnancy-related health and current medications. The information will be collected onto standardised case report forms (eg, see online supplementary appendix 4). Where required, our definitions of variables included in the 4P study are detailed in online supplementary appendix 5.

At the discharge appointment, participants will be asked to fill out the standardised system usability survey[25] to assess the usability of the 4P mobile system for home monitoring (online supplementary appendix 6).

### Follow-up

In the antenatal period, the participants will be followed up at four to six weekly intervals. If they are coenrolled in the Interbio-21st study, antenatal data will be collected by Interbio-21st staff. During the immediate postpartum period (days 0–14 after delivery), participants will be asked to undertake daily home monitoring. Additional postnatal data will be collected by research midwives on two to three occasions. If participants do not respond to contact by research midwives during the study, they will be treated as withdrawn after the third failed contact attempt. Data already collected from withdrawn participants will be included in the final analysis, unless explicitly requested otherwise. If participants complete the study they will receive a £20 gift voucher. A participant's timeline for the study is shown in figure 2.

# Timeline for data collection

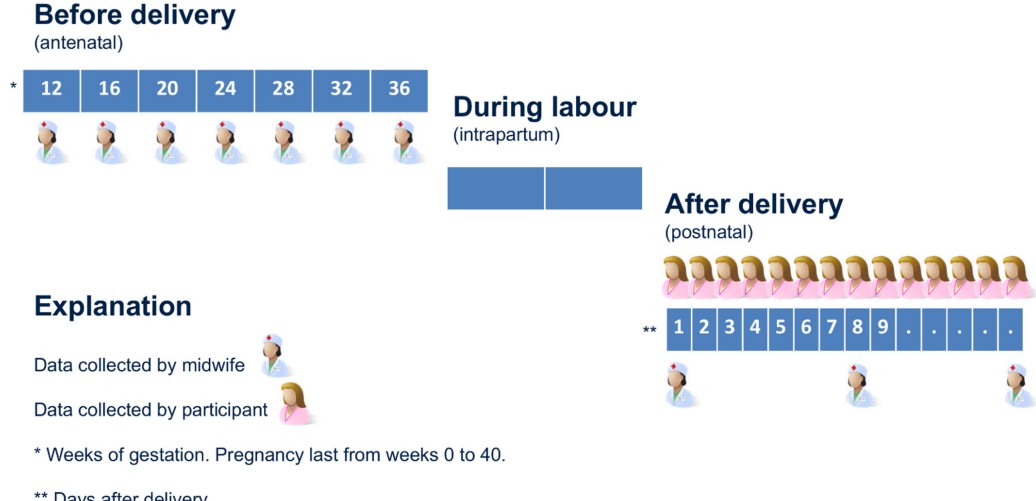

**Figure 2** Timeline of follow-up visits during the antenatal and postnatal periods.

## Personnel

Trained research midwives will undertake all antenatal observation sets, and one to three postnatal observation sets, following study SOPs (eg, see online supplementary appendix 7). Research midwives will provide training in the use of home monitoring equipment at a suitable time around delivery, as described by SOPs. If abnormal vital signs are identified at the home visits, the research midwife will refer the participant to her usual midwife, general practitioner or local maternity hospital assessment unit. All participants will be made aware that the information collected during the home monitoring phase is not reviewed in real time and that it remains their responsibility to seek assistance if they feel unwell. The research midwives will extract all hospital data from the notes and observation charts into an electronic study database, following study SOPs. Research midwives will perform frequent site visits to carry out midwife training and address any recruitment and equipment issues.

## Sample size

In order to determine the sample size we considered the precision of the data in estimating specific centiles. We assumed data are normally distributed at each gestational age. The SE of the $P^{th}$ centile is given using the standard formula[26]:

$$SE_P = SD\sqrt{\left(1 + \tfrac{1}{2}Z_P^2\right)/n}$$

where SE is the standard error, SD is the standard deviation of the measurement of interest (which may change according to gestational age), $Z_p$ is the value of the standard normal distribution corresponding to the $P^{th}$ centile, and n is the sample size. To create an evidence-based early warning score we desired a 95% CI with an SE of less than 0.10*SD at the boundaries. A sample size of 1000 women will achieve an SE of 0.05*SD at the 2.5th and 97.5th centiles, and even greater precision at the less extreme centiles. Adequate precision is also met for any subgroup analysis; for example, a sample size of 300 women will achieve an SE of 0.1*SD at the 2.5th and 97.5th centiles. We have been conservative in these estimates as we have not considered the effect of serial measurements from the same women.[27] Using Royston's design factor of 2.3, a longitudinal study of 1000 women could have equivalent precision to a cross-sectional study of around 2300 women.[28]

## Data management

A custom-designed website will be the main portal for manual data entry and vital sign review (figure 3). Access will be restricted by username and 8-digit password to staff directly involved in the study. The 4P database and interfaces will be held within a secure virtual server protected with an NHS network firewall. For manual chart entry, automatic checks will be performed to ensure sequential chart entry and reasonable vital sign values.

## Data quality

Duplicate measurements will be recorded for a subset of participant visits to document the effect of using only the first measurement (as in clinical practice). To minimise participant inconvenience, duplicate measurements will be reviewed by study statisticians to decide when an adequate number of duplicate measurements have been collected. Research midwives will undertake observation sets in the home to allow comparison between participant-taken and midwife-taken observations.

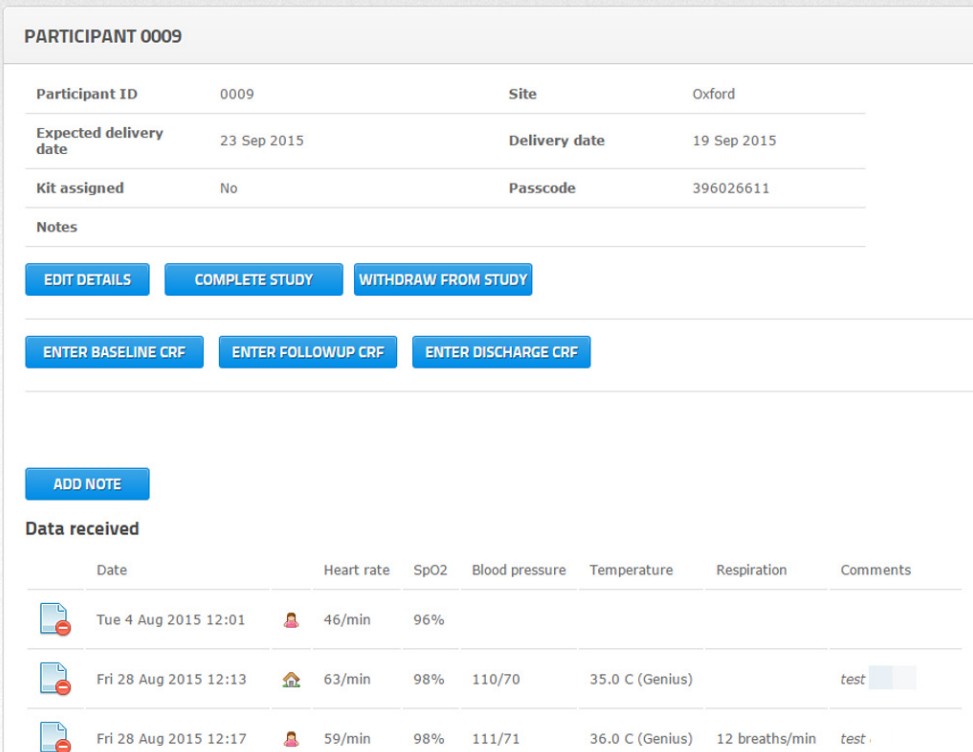

**Figure 3** Review of vital sign data for participant 0009 on the 4P website (example).

Temperature will be measured with both Bluetooth-enabled and non-Bluetooth-enabled devices to allow method comparison. Quality of the pulse oximeter recordings (heart rate and oxygen saturation) will be assessed in real time by observation of the photoplethysmograph waveform (only visible to midwives). Additionally, the electronic heart rate measurement will be checked with a 60 s pulse palpation.[29]

Vital sign data retrospectively extracted from patients' notes will be directly entered into the electronic case report form. The form automatically charts the vital signs onto electronic charts similar in appearance to the paper chart as the vital signs are entered, facilitating reliable data capture. Dual entry will be undertaken in 10% of cases to assess reliability.

### Data analysis

Statistical methods to generate gestational age-specific centiles for maternal vital signs will closely follow those methods described and used in the INTERGROWTH-21st Project for fetal growth measurements.[30] Predefined exclusion criteria will be applied to make sure only those with a normal pregnancy are included in the analysis. We will explore different statistical methods in order to achieve the best fit to the data, while being as simple as possible. In brief, these methods will include: mean and SD using fractional polynomials[31]; Cole's lambda, mu and sigma method[32]; the lambda, mu, sigma and Box-Cox t distribution method[33]; the lambda, mu, sigma and Box-Cox power exponential method[34]; and multilevel models. To present curves,

we will use different smoothing techniques including fractional polynomial,[34] cubic splines[35] and penalised splines.[36] The INTERGROWTH-21st Project concluded that there was no evidence of a non-normal distribution and therefore no benefit of the more complex modelling techniques and opted to use the mean and SD method.[30] We therefore anticipate that values of each vital sign will be normally distributed at each stage of pregnancy, and that we will also be able to choose the simpler mean and SD method as the best option. In this case, we will build the model in a multilevel framework, taking account of the correlation in repeated measures within participants.

Goodness of fit will be assessed by visual inspection of empirical centiles versus fitted centiles, quantile-quantile plots of the residuals, plots of residuals versus fitted values, and the distribution of fitted Z scores across gestational ages.

It is expected that some participants will become lost to follow-up or will have some missing measurements. These participants will still be included in the analysis, unless all data are missing. Missing data will be implicitly imputed using the multilevel modelling techniques under the missing at random assumption.

Centiles will be derived in the following predefined subgroups: body mass index, maternal age, parity, smoking status and ethnicity, and will be visually compared with assessed differences between subgroup strata. Sensitivity analyses will be performed to assess the robustness of the final model to the various assumptions made, including

assumptions around missing data, and the choice of participants included in the analysis.

## Study timelines

Expected study timelines are shown (online supplementary appendix 8).

## ETHICS AND DISSEMINATION
### Ethical considerations

Informed written consent will be sought for all participants and participants can withdraw from the study at any time. The study is non-invasive and presents no significant risk to participants. Data are managed in accordance with the Data Protection Act (1998).

Participants in the 4P study will be observed more frequently than they would be as per usual maternity care. However, the opportunity to meet with health professionals more frequently could be considered beneficial. During home visits, either antenatally or postnatally, the research midwife will refer participants to their usual care team if any abnormal vital sign values are recorded. Participants are asked to monitor their own vital signs at home. There is the possibility that some measurements are abnormal. If elevated blood pressure values are recorded, the software on the Android tablet advises study participants to contact their local maternity unit (a list of telephone numbers is provided). Participants are advised that the research team does not monitor the data sent from home and that they are responsible for acting on the software instructions whenever any abnormally high or low values of vital signs are recorded.

### Dissemination

Results from the study will be disseminated through conferences and published in international peer-reviewed journals. Participants will be able to access results through a link on the OSPREA (Oxford Safer Pregnancy Alliance) website. Important protocol modifications will be communicated on the ISRCTN where the study is registered (No. 10838017). The data will be analysed by the University of Oxford (Institute of Biomedical Engineering) in the first instance. Anonymised data will be made available to other researchers on application.

## CONCLUSIONS

An evidence-based early warning system will enable clinicians to identify more quickly the ill or deteriorating woman in a hospital setting at any time during pregnancy, delivery and the immediate postpartum period. Currently there are a number of EWS charts for pregnant women being used in the UK with different thresholds for alerts, none of which are evidence based. The 4P study will enable a standardised evidence-based obstetric EWS chart to be developed from a database of vital signs for gestation of pregnancy, delivery and 2 weeks postpartum.

**Author affiliations**
[1]Nuffield Department of Obstetrics and Gynaecology, Oxford University Hospitals NHS Foundation Trust, Oxford, UK
[2]Department of Engineering Science, University of Oxford, Oxford, UK
[3]Nuffield Department of Orthopaedics, Rheumatology and Musculoskeletal Sciences, Centre for Statistics in Medicine, Oxford, UK
[4]Musculoskeletal Sciences, University of Oxford, Oxford, UK
[5]Nuffield Department of Clinical Neurosciences, University of Oxford, Oxford, UK

**Contributors** PJW is the principal investigator. PJW, LHM, LL, LT, OG, JK and FK made substantial contributions to the conception and design of the project. SG provided advice on statistical methodology. JK, FK and CE are involved in data collection. OG, AT, DS, LL and RMP are involved in software design and data analysis. FK wrote the first draft of the present manuscript. RMP and PJW reviewed and updated the draft. All authors read and approved the final manuscript.

**Funding** The 4P study is supported by the NIHR Biomedical Research Centre, Oxford. RMP and LL acknowledge the support of the RCUK Digital Economy Programme grant number EP/G036861/1 (Oxford Centre for Doctoral Training in Healthcare Innovation). LL is also supported by the Clarendon Fund. University of Oxford. Contact Heather House, Joint Research Office, Block 60, Churchill Hospital, Old Road, Oxford OX3 7LE.

**Competing interests** None declared.

**Patient consent** Detail has been removed from this case description/these case descriptions to ensure anonymity. The editors and reviewers have seen the detailed information available and are satisfied that the information backs up the case the authors are making.

**Ethics approval** NRES committee South East Coast-Brighton and Sussex (Ref: 14/LO/1312)

**Provenance and peer review** Not commissioned; externally peer reviewed.

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
