## [Reviewer comments · BMJ Open]

ARTICLE DETAILS

TITLE (PROVISIONAL)	The Pregnancy Physiology Pattern Prediction Study (4P Study): protocol of an observational cohort study collecting vital sign information to inform the development of an accurate centile-based obstetric early warning score
AUTHORS	Kumar, Fiona; Kemp, Jude; Edwards, Clare; Pullon, Rebecca; Loerup, Lise; Triantafyllidis, Andreas; Salvi, Dario; Gibson, Oliver; Gerry, Stephen; MacKillop, Lucy; Tarassenko, Lionel; Watkinson, Peter

VERSION 1 - REVIEW

REVIEWER	Professor Michael Turner Professor of Obstetrics and Gynaecology Director UCD Centre for Human Reproduction Coombe Women and Infants University Hospital
REVIEW RETURNED	20-Feb-2017

GENERAL COMMENTS	This is an important and topical issue in contemporary maternity care. I was confused by the protocols. It states the study will start in August 2012. This protocol is dated January 2017 and has 11 authors. There is also a second protocol dated 14 September 2015 with four of the 11 authors. The last references are 2014 which may explain why the text appears a little dated and may explain the comment that current obstetric EWS do not reflect the physiological changes of pregnancy. This statement does not hold true for the obstetric EWS introduced nationally in both the north and south of Ireland. I accept however that there is scant literature historically on the measurements of vital signs as pregnancy advances. I am also puzzled by the decision to base normality and abnormality on centiles. What is the basis for this? Surely, the observations need also to be linked to different obstetric or medical complications. Is there a risk that observations by a midwife may impact on the measurements? I am surprised there is not more discussion about wearable devices and I suspect there may have been significant developments since this protocol was first written. I also note that there is a plan to copy measurements retrospectively from the clinical records. This runs the risk of errors and of measurements being rounded up or down. Again, this could be overcome with the use of wearable devices that could be used as both inpatients and outpatients.
---

REVIEWER	Laurence Shields, MD Dignity Health
REVIEW RETURNED	10-Mar-2017

GENERAL COMMENTS	The idea of collecting vital sign data for pregnancy as noted is not new but the context of then using contemporary data to create data driven parameters for early warning criteria is novel and needed. The abstract is confusing as it states " we will start in august 2012" Is this an error ? Can the authors be more specific the position that the patient will be in when the various parameters are collected and how this may change during the course of the pregnancy. For BP measurements they state that they will only use the first measurement, this is frequently elevated and most recommend repeat -- how will this be addressed?
---

VERSION 1 – AUTHOR RESPONSE

Reviewer: 1

Professor Michael Turner

Professor of Obstetrics and Gynaecology

Director UCD Centre for Human Reproduction

Coombe Women and Infants University Hospital

Please state any competing interests or state 'None declared': None declared

This is an important and topical issue in contemporary maternity care.

I was confused by the protocols. It states the study will start in August 2012. This protocol is dated January 2017 and has 11 authors. There is also a second protocol dated 14 September 2015 with four of the 11 authors.

We apologise for causing confusion. The first stage of the 4P study occurred as a substudy of the Interbio 21-st study, commencing 1 August 2012. The second stage allowed the study to expand to two other sites, and allowed the first site (Oxford) to continue after the InterBio-21st study completed. As a result the first stage was done under an ethics amendment to the InterBio-21st study, the second with a separate ethics application and protocol. We have clarified this both in the text (Methods and analysis – study design) and in appendix 8 – study timelines – as suggested by the editors. Since the study began it has become accepted good practice to publish the protocol before data analysis begins. This is what we wish to do. The protocol submitted for publication in BMJ open encompasses the whole 4P study, including patients recruited as a subset of Interbio-21st. We originally attached the latest version of the protocol submitted for ethical approval for stage 2 to our original submission. In retrospect this was both confusing for the reader and an over-interpretation of BMJ open guidance. We have therefore removed this from the current submission.

The last references are 2014 which may explain why the text appears a little dated and may explain the comment that current obstetric EWS do not reflect the physiological changes of pregnancy. This statement does not hold true for the obstetric EWS introduced nationally in both the north and south of Ireland. I accept however that there is scant literature historically on the measurements of vital signs as pregnancy advances.

We believe it is important to explain the basis for doing the 4P study, so we have not updated references beyond those relevant when we commenced stage 2.

Our abstract states: "nor do they reflect the changing physiology that occurs with gestation during pregnancy". We note that the I-MEWS (introduced in Ireland) makes no alteration for different gestations.

I am also puzzled by the decision to base normality and abnormality on centiles. What is the basis for this? Surely, the observations need also to be linked to different obstetric or medical complications.

We have addressed this issue in paragraph 4 of the introduction and referenced our (both highly cited and used in clinical practice) paper using the centile-based approach. The approach is particularly suited to pregnancy, where event rates are low. Further explanation of the benefits of this approach is contained within this paper.

Is there a risk that observations by a midwife may impact on the measurements?

Vital signs as taken by healthcare professionals such as midwives are those used for early warning. Normal ranges obtained by midwives are therefore what is required. The reviewer is correct that this may mean the vital signs taken by participants in the post-partum period may require adjustment. It is for this reason that midwife measurements will also be taken post-partum.

I am surprised there is not more discussion about wearable devices and I suspect there may have been significant developments since this protocol was first written.

The aim of the 4P study is to provide evidence-based early warning using the vital signs taken in normal practice, using normal equipment. We have undertaken studies involving wearable devices over many years (for example Clifton DA, *BMJ Open* 2015;5:e00737, Villarroel M, *Healthcare Technology Letters* 2014;3:87–91, Clifton L, *IEEE Journal of Biomedical and Health Informatics* 2014;18(3):722-730, Bonnici T, *Clin Med.* 2013;13(3):252-7, Pimental M, *Med Biol Eng Comput.* 2013;51(8):869-77, Clifton L, *IEEE Transactions on Biomedical Engineering* 2013;60(1):193-7). We chose the devices for women in to use in the home in light of this expertise. We think it will be some time before wearable devices are in routine use in pregnant women.

I also note that there is a plan to copy measurements retrospectively from the clinical records. This runs the risk of errors and of measurements being rounded up or down. Again, this could be overcome with the use of wearable devices that could be used as both inpatients and outpatients.

We have developed electronic systems to minimize the risk of data entry errors – examples of these are included in the current submission (appendix 3). We have developed these systems in light of our experience in previous studies with wearable monitoring (Clifton DA, *BMJ Open* 2015;5:e00737, Jeffs E, *J Adv Nurs.* 2016 Aug;72(8):1851-62). We will also double data enter at least 10% of entries to check for accuracy. We will present the results of this analysis in the paper. We have clarified this in the protocol (data quality para 3).

Reviewer: 2

Laurence Shields, MD

Dignity Health

Please state any competing interests or state 'None declared': None Declared

The idea of collecting vital sign data for pregnancy as noted is not new but the context of then using contemporary data to create data driven parameters for early warning criteria is novel and needed.

The abstract is confusing as it states " we will start in august 2012" Is this an error ?

We have changed the tense to say that the study commenced in 2012 – again this is a function of us

nearing study completion but wishing to publish the protocol before it is possible to commence data analysis – as is now considered good practice.

Can the authors be more specific the position that the patient will be in when the various parameters are collected and how this may change during the course of the pregnancy.

These are contained within Appendix 7 – standard operating procedures.

For BP measurements they state that they will only use the first measurement, this is frequently elevated and most recommend repeat -- how will this be addressed?

We need to establish the range of blood pressures found in normal practice on first measurement – not what the blood pressure may adjust to on repeat However, our methods will allow us to estimate any effect of repeated measurement (see data quality para 1).